# Adenomyosis—A Call for Awareness, Early Detection, and Effective Treatment Strategies: A Narrative Review

**DOI:** 10.3390/healthcare12161641

**Published:** 2024-08-17

**Authors:** Georgios Kolovos, Ioannis Dedes, Sara Imboden, Michael Mueller

**Affiliations:** Department of Obstetrics and Gynecology, University Hospital of Bern, University of Bern, 3010 Bern, Switzerland; ioannis.dedes@insel.ch (I.D.); sara.imboden@insel.ch (S.I.); michel.mueller@insel.ch (M.M.)

**Keywords:** adenomyosis, reproductive health, early detection, minimally invasive therapy

## Abstract

Objective: To provide a brief summary of the high incidence, symptomatology, different types, and diagnosis of adenomyosis and to explore various aspects of the disease, with the primary aim of raising awareness among gynecologists for appropriate and early detection. Background: Adenomyosis, a benign gynecological condition characterized by the infiltration of endometrial tissue into the myometrium, poses significant challenges to women’s reproductive health. Methods: A narrative review was conducted by searching PubMed, Scopus, and Cochrane databases and offering a non-systematic summary and critical analysis of current knowledge on the impact of adenomyosis on women’s health. Articles published in the English language up to May 2023, including original scientific papers, clinical trials, meta-analyses, and reviews focusing on various aspects of adenomyosis, were included in the synthesis of this review. Conclusions: Approximately 20% of women are affected by adenomyosis, which manifests with various subtypes, distinct epidemiological profiles, symptomatology, and treatment responses. Despite its clinical significance, adenomyosis remains understudied, resulting in a significant disparity in research and the literature compared to other gynecological conditions. The severity of adenomyosis is compounded when coexisting with endometriosis, particularly deep-infiltrating endometriosis (DIE), leading to exacerbated fertility issues and severe symptomatology. The wide range of symptoms, including adverse pregnancy outcomes such as pre-eclampsia, highlights its wider impact and emphasizes the need for increased awareness of the condition. Adenomyosis is frequently associated with treatment failure in endometriosis, contributing to dienogest resistance, elevated discontinuation rates, and persistent pain post-endometriosis surgery. Additionally, the lack of specific treatments tailored to adenomyosis poses a considerable challenge in clinical management.

## 1. Introduction

Adenomyosis is a prevalent yet often underrecognized gynecological condition involving the invasion of endometrial tissue into the myometrium. This condition significantly impacts women’s reproductive health, contributing to a wide array of symptoms and complications. Despite its high incidence and clinical importance, adenomyosis does not receive adequate clinical attention and research compared to other gynecological disorders. The aim of this narrative review is to consolidate current knowledge about adenomyosis, focusing on its incidence, subtypes, diagnostic techniques, symptomatology and complications, co-occurrence with endometriosis and resistance to treatment in endometriosis patients as well as the lack of specific treatment modalities. By doing so, we aim to enhance awareness among healthcare providers and encourage early and accurate diagnosis. This article is presented in accordance with the narrative review reporting checklist (available at https://gpm.amegroups.org/article/view/10.21037/gpm-20-57/rc, accessed on 15 May 2024) [1].

## 2. Methods

We conducted a thorough literature search on PubMed, Scopus, and Cochrane databases using MeSH or index terms for the keyword: “adenomyosis”. We included articles published in the English language from the last thirty years, from May 1994 to May 2024, from a diverse range of sources such as original scientific papers, clinical trials, meta-analyses, and reviews. We found 3625 articles from PubMed, 2170 from Scopus, and 194 from Cochrane, and 43 articles were selected by the authors and additional references were also searched from the bibliographies of those chosen articles.

## 3. Pathogenesis

Sampson was a pioneering figure in the classification and understanding of adenomyosis, initially describing the condition as ‘adenomyoma’ [2]. He proposed three potential pathways for the development of adenomyomas: invasion of normal endometrium into the myometrium, invasion from extrinsic sources such as endometrial cysts, or misplacement of endometrial tissue within the uterine wall. Over time, various pathophysiological theories have emerged to explain the formation of adenomyosis. These include microtrauma theory, which focuses on tissue injury and repair mechanisms at the endometrial–myometrial interface; the theory of invasion of endometrial basalis cells into the myometrium; the de novo metaplasia or Mullerian remnant theory; and the outside to inside theory. Each of these theories offers insights into the complex mechanisms underlying the development of adenomyosis, contributing to our evolving understanding of this enigmatic condition. 

Going through the most prominent theory, the mechanism of tissue injury and repair (TIAR) plays a crucial role. This suggest that chronic peristaltic contractions of the myometrium may lead to auto-traumatization of the predisposed myometrium, potentially initiated by traumatized endometrial–myometrial interfaces, which are often observed following uterine manipulating procedures like hysteroscopy or cesarean section. This chronic peristalsis induces chronic microtrauma and inflammation, releasing cytokines and promoting increased estrogen production. This hyperestrogenic environment may lead to invagination of the endometrial basal layer into the uterine muscular layer, forming adenomyotic lesions. 

Hence, a vicious cycle is reinforced, where hyperperistalsis in the junctional zone induces repeated auto-traumatization cycles where the constant disruption of the myometrial muscular fibers and the fibroblasts at the endometrial–myometrial connection leads to an invagination of the endometrial basal layer to the muscular layer of the uterus [3]. While the prevailing theory suggests that adenomyosis arises from the invasion of endometrial tissue into the myometrium, certain cases challenge this hypothesis. For instance, in patients with Mayer–Rokitansky–Kuster–Hauser syndrome, where functional endometrium is absent, adenomyosis has been histologically confirmed, suggesting alternative mechanisms. Thus, it is proposed that adenomyosis may develop through metaplastic transformation of embryonic epithelial progenitors or differentiation of adult endometrial stem cells migrating to and integrating within the myometrium [4]. Additionally, in cases lacking direct histological evidence linking adenomyosis to the endometrium, the conventional TIAR mechanism may not fully apply. Instead, the pathology may originate from the invasion of endometrium-like structures from outside the uterus, disrupting the uterine serosa.

## 4. Symptomatology

The main symptoms of adenomyosis are pain (e.g., dysmenorrhea and pelvic pain) and abnormal uterine bleeding (AUB). Accompanying diseases with comparable symptomatology are often encountered such as uterine fibroids and endometriosis, making it challenging to differentiate which pathology is accountable for individual symptoms. 

Rates of dysmenorrhea among patients with adenomyosis range from 30% to 68% [5,6]. Patients with adenomyosis may often present with deep dyspareunia, often mimicking the dyspareunia of patients with deep-infiltrating endometriosis. The pathophysiology of the pain is not fully understood but chronic inflammation and prostaglandins may play an important role [7]. Regarding AUB, the International Federation of Gynecology and Obstetrics (FIGO) recognizes adenomyosis as a distinct entity within the PALM- COEIN classification system. Uterine adenomyosis is a potential cause of abnormal uterine bleeding, occurring in 20% to 35% of cases of AUB [8]. This classification system categorizes causes of AUB into specific groups, with “A” representing adenomyosis. Natalin et al. reported a significant increase in menstrual blood loss with an increase in the disease burden [9]. To further complicate things, about one-third of patients with adenomyosis have no symptoms [10]. 

## 5. Key Reasons for Increasing Awareness of Adenomyosis


**High Incidence:**


Historically, adenomyosis diagnosis relied primarily on histological examination post-hysterectomy, which often underestimated the condition’s prevalence and significance. With advancements in imaging technologies like transvaginal ultrasound and MRI, diagnosis has significantly improved, revealing a broader demographic of affected individuals, particularly younger women of reproductive age. This shift has highlighted a higher than previously recognized prevalence of adenomyosis, estimated to affect 15–20% [11,12] of young women, often coexisting with endometriosis and uterine fibroids. Similarly, in cases where hysterectomies are performed for urogynecological reasons, the prevalence ranges from 20% to 30% [13,14]. In women suffering from dysmenorrhea and abnormal uterine bleeding, prevalence rates can be notably higher, ranging from 30% to 59%, as reported in some studies [15,16]. Notably, 40% of women diagnosed with endometriosis also have adenomyosis present, while 80% of infertile women diagnosed with endometriosis have also adenomyosis, with over 40% of these cases involving deep-infiltrating endometriosis (DIE) lesions [17]. In women undergoing assisted reproductive technologies (ARTs), adenomyosis is prevalent in approximately 20% to 25% of cases [18]. Increased recognition is crucial for appropriate diagnostic and management approaches;


**Disparity in the Research and the Literature:**


Despite its clinical significance, adenomyosis remains understudied compared to endometriosis, as evidenced by the disparity in the scientific literature, with only 3982 entries for adenomyosis versus 35,132 for endometriosis on PubMed (as of 1 May 2024) Historically, even comprehensive guidelines for endometriosis seldom addressed adenomyosis, reflecting an oversight in clinical protocols. The recent emergence of guidelines on adenomyosis such as from the Asian Society of Endometriosis and Adenomyosis and the Society of Obstetricians and Gynecologists of Canada (SOGC) [19,20] mark the progress in standardizing the management and treatment of adenomyosis. Furthermore, efforts are underway to systematically address the research gap, such as the “*Development of a core outcome set and outcome definitions for studies on uterus-sparing treatments of adenomyosis (COSAR): an international multistakeholder-modified Delphi consensus study*”. This initiative aims to standardize research outcomes to enhance the quality and comparability of studies on adenomyosis;


**Different Types and Classifications of Adenomyosis:**


Recent advancements in imaging technologies, such as transvaginal ultrasound and magnetic resonance imaging, have significantly transformed the landscape of adenomyosis diagnosis and also reveal different patterns of adenomyosis: most classification models typically delineate between focal and disseminated disease, as well as adenomyosis of the inner and outer myometrium. While various classification systems are available, one of the most comprehensive and widely utilized models is the one proposed by Kishi et al. [21]. In this pioneering work, Kishi et al. introduced a comprehensive classification system for uterine adenomyosis based on MRI analysis. Recognizing the two primary forms of adenomyosis, diffuse and focal, and their dual localizations relative to the junctional zone, as initially reported by Kishi et al., is essential. The extrinsic form of adenomyosis typically spares inner structures such as the endometrium and junctional zone but often disrupts the serosa; in contrast, the intrinsic form affects the inner structures, underscoring the importance of distinguishing between these manifestations for a comprehensive understanding of adenomyosis pathology and symptomatology.

Emerging data indicate that different subtypes of adenomyosis—especially intrinsic versus extrinsic and diffuse versus focal—have different etiologies and clinical profiles. Intrinsic adenomyosis, more common in older patients and often associated with abnormal uterine bleeding (AUB) and prior uterine surgery, contrasts with extrinsic adenomyosis, which is typically found in younger, nulligravida women and is closely associated with deep-infiltrating endometriosis and primary infertility [22,23]. 

In 2015, the international Morphological Uterus Sonographic Assessment (MUSA) group issued a consensus regarding the appropriate terminology for describing myometrial lesions observed on ultrasonography [24]; 


**Compounded Severity with Endometriosis:**


Chapron et al. in 2017 introduced the “outside to inside invasion” theory, proposing that ectopic endometrial cells migrate from posterior endometriosis nodules into the myometrium. Their 2017 prospective observational study revealed a statistically significant correlation between focal adenomyosis of the outer myometrium (FOAM) and deep-infiltrating endometriosis phenotypes, with a co-occurrence rate of 66.3% (110 cases; *p* < 0.001). Conversely, the co-occurrence rates with superficial peritoneal endometriosis and endometriomas were lower, at 7.5% (3 cases) and 19.3%, respectively [25]. Parker et al. revealed a significant association between adenomyosis and specific sites affected by endometriosis, particularly deep retrocervical endometriosis (60%; *p* = 0.01) and involvement of the rectosigmoid (49.2%; *p* = 0.03) [26]. 

The same study group in a prospective observational study involving 255 symptomatic deep-infiltrating endometriosis patients discovered a prevalence of 56.5% of focal adenomyosis of the outer myometrium (FOAM). The prevalence of multiple DIE lesions was found to be notably higher in the FAOM-positive group compared to the FAOM-negative group. Specifically, among the FAOM(+) participants, 82.6% (119 out of 144) exhibited multiple lesions, while in the FAOM(−) group the prevalence was 58.6% (65 out of 111), indicating a strong association between the presence of FAOM and the increased likelihood of multiple DIE lesions [27]. The presence of adenomyosis in patients with endometriosis often results in more severe symptoms, complicating the management of endometriosis and necessitating integrated therapeutic strategies that address both conditions;


**Association with Treatment Failure in Endometriosis:**


Adenomyosis is a major cause of persistent pelvic pain and can lead to higher rates of treatment failure, both surgically and medically, especially in patients being treated for endometriosis [28,29]. This highlights the necessity for clinicians to consider adenomyosis in their differential diagnoses after endometriosis treatment, as well as in managing ongoing symptoms. Medical therapies, such as dienogest used for treating endometriosis, may be less effective when adenomyosis is also present, as high discontinuation rates are reported due to persistent bleeding [22,30]. The often limited effectiveness of progesterone-based therapies in adenomyosis involve progesterone resistance, which stems from a reduction in the expression of PGR, particularly the PGR-B isoform [31,32]. The presence of adenomyosis has also been established as an independent risk factor for complications in deep endometriosis laparoscopic surgery [33]; 


**Lack of Specific Treatments:**


There are currently no drugs specifically labeled for the treatment of adenomyosis, which complicates management strategies and highlights the need for innovative therapeutic development and tailored treatment approaches. Medical treatment serves as the primary option for patients with adenomyosis who aim to preserve fertility or are advised for individuals ineligible for surgery due to concurrent medical conditions because of the high morbidity and recurrence rates [34]. Regarding medical therapies, potential treatment options encompass combined oral contraceptive (COC) pills, progestins, the levonorgestrel-releasing intrauterine system (LNG-IUS), and gonadotropin-releasing hormone (GnRH) agonists and antagonists [35]. The effectiveness of these therapies varies, with the LNG- IUS being the most effective—although in off-label use—in reducing the pain and the abnormal uterine bleeding by inducing decasualization and atrophy of the endometrium and down-regulating estrogen receptors through increased continuous local progesterone release [36]. 

As a second step for patients with treatment resistance or severe symptoms and in cases where organ preservation is warranted, a range of interventional procedures are available, including uterine artery embolization (UAE) and a broad category of hyperthermic treatments. Hyperthermic treatments encompass procedures utilizing energy sources such as high-intensity focused ultrasound (HIFU), percutaneous microwave ablation (PMWA), and radiofrequency ablation (RFA). Two recent systematic reviews and meta-analyses showed that hyperthermic therapy including RFA may be effective and safe minimally invasive therapies for symptomatic adenomyosis [37,38].

Other uterine-sparing surgeries include uterine artery ligation and adenomyosis excision. Surgical resection of diffuse and focal adenomyosis, excluding adenomyomas, demands significant surgical expertise and is associated with notable perioperative risks. The last treatment option for severe adenomyosis and completed family planning remains hysterectomy; 


**Adenomyosis is linked to infertility and negative outcomes during pregnancy.**


The study of Mishra et al. [39] was the first systematic review to evaluate the prevalence of both isolated adenomyosis and adenomyosis with coexisting endometriosis and/or fibroids in women with subfertility. The pooled prevalence was 10% for isolated adenomyosis, 1% for adenomyosis with coexisting fibroids, 6% for adenomyosis with coexisting endometriosis, and 7% for adenomyosis with endometriosis. The answer remains unclear for which type of adenomyosis has the worst fertility outcome. There is some recent evidence, though, that suggests that the focal type of the disease might have a more significant negative effect than other forms but more evidence is definitely needed [40].

Beyond its impact on pregnancy rates, adenomyosis has been linked to negative obstetric outcomes. In a recent systematic review and metanalysis, Nirgianakis et al. reported that following treatment with assisted reproductive technology (ART), there was a notably higher miscarriage rate (odds ratio [OR] 2.17; 95% confidence interval [CI] 1.25–3.79) when adenomyosis was present. Furthermore, adenomyosis was further found to be significantly linked with an elevated risk of adverse obstetric outcomes, including pre-eclampsia. Τhese findings were also confirmed by the studies of Horton et al. and Razavi et al., with ORs ranging from 4.35 to 7.87 [41,42,43]. 

Several other complications seem also to be increased in patients with adenomyosis including an increased prevalence of small for gestational age infants, preterm delivery [41,42,43,44], caesarean section [41,43], fetal malpresentation [42,43], and postpartum hemorrhage [43]. These associations remained significant even after adjustments for age and mode of conception were made, highlighting the intrinsic risks associated with the presence of adenomyosis in pregnancy. Last but not least, women with adenomyosis situated on the posterior side are identified to experience severe obstetric complications, including placenta previa, placenta accreta, preeclampsia, and preterm birth according to Shi et al. [45]. These data emphasize the importance of careful monitoring and management of pregnant women diagnosed with adenomyosis, to mitigate these risks and improve pregnancy outcomes [43].

## 6. Conclusions

This narrative review has several limitations that have to be acknowledged. The non-systematic methodology used in this review does not follow a standardized protocol for article selection and data extraction, potentially leading to inconsistent inclusion criteria, incomplete coverage, and variable study quality. This review and commentary offer an initial framework for assessing the impact of adenomyosis on women’s health and underscores the need for awareness of the disease, early detection, and improved treatment approaches for this often overlooked condition. However, several critical research questions persist. 

## 7. Call for Awareness

With its profound impact on the quality of life of affected women, it is imperative that healthcare professionals, researchers, and policymakers unite to address the challenges posed by adenomyosis. By raising awareness among both healthcare providers and the general public, we can ensure timely diagnosis and intervention, ultimately improving outcomes and quality of life for affected individuals. Early detection through improved diagnostic techniques and screening protocols is essential for timely intervention and preventing disease progression. Moreover, concerted efforts are needed to develop and implement more effective treatment strategies tailored to individual patient needs. Through collaborative research and education, we can strive towards better recognition, understanding, and management of adenomyosis, ultimately enhancing the well-being of the affected women worldwide.

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
