# Peer review of "Adenomyosis—A Call for Awareness, Early Detection, and Effective Treatment Strategies: A Narrative Review"

_healthcare, 2024, doi:10.3390/healthcare12161641_

Round 1
Reviewer 1 Report
Comments and Suggestions for Authors
The authors used published literature to develop the manuscript on a very interesting subject, with a notable percentage of women (20%) facing this issue. However, the major issue with the manuscript lies in the methodology. While it appears to be a literature review, some clarifications are needed. I encourage the authors to add more details to the methodology and address the following questions:
- What time period was covered in the literature review? For example, was it from 2010 to 2023?
- How were the articles selected for this review?
- Did the authors include literature from all countries? Did they include any book chapters or grey literature?
- Did the authors include publications in English only or all types of publications?
- Which approach was used for this review? Was it a deep literature review or a scoping review? If it was a scoping review, the elements of a scoping review must be addressed. If it was a literature review, the authors should clarify the above questions and also add a limitations section to discuss potential limitations.
The MS benefit from the structural MS, to add introduction, Methods, Findings, Discussion and Conclusions.
The MS needs major revision.
Author Response
Dear Sir or Madam
I would like to thank you for your comments on our manuscript.
You are absolutely right about the incomplete methodological presentation of the article, which you can now find updated on the lines 16-20 on the abstract, as well as on lines 44- 52 of the final MS. We carried out a narrative review, which was completed according to the following checklist ( https://gpm.amegroups.org/art cle/view/10.21037/gpm-20-57/rc).
We includes all possible articles that where identified on the PubMed, Scopus and Sciencedirect from publication up to May 2024 (lines 49-51)
We includes diverse range of sources such clinical trials, meta-analyses and reviews (lines 51-52)
We only included articles published in English language (line 50-51)
Potential limitations are mentioned on the conclusion (lines 266-269)
The MS has now been completely revised to be methodologically correct.
Best regards
Georgios Kolovos

Reviewer 2 Report
Comments and Suggestions for Authors
A fairly reasonable review paper about adenomyosis from the pathogenesis, symptomatology, and ways to increase awareness etc....... It might get some interest from the reader to read. Most of the references are cited without the year of publication and are not standardized. Review at each session/subtopics are quite superficial and not extensive.
eg. Regarding medical therapies, potential treatment options encompass combined oral contraceptive (COC) pills, progestins, the levonorgestrel-releasing intrauterine system (LNG-IUS), and gonadotropin-releasing hormone (GnRH) agonists and antagonists. This is a very general statement and does not cite the reference or evidence.
Lastly, the assessment above by the journal is for research and not a review paper.
Author Response
Dear Sir or Madam
I would like to thank you for your comments on our manuscript. You are right about the incomplete methodological presentation of the article, which you can now find updated. We carried out a narrative review, which was completed according to the following checklist (https://gpm.amegroups.org/art cle/view/10.21037/gpm-20-57/rc). The aim of this review was to present a brief summary of various aspects of adenomyosis with the aim of raising awareness and providing information on these topics, which are often unknown to many gynaecologists. Each topic is not too extensive, as the subject of adenomyosis is so broad that the article would have been extremely long.
As far as the medical therapies are concerned we added the citation (line 215); Vannuccini S, Luisi S, Tosti C, Sorbi F, Petraglia F. Role of medical therapy in the management of uterine adenomyosis. Fertil Steril. 2018 Mar;109(3):398-405. doi: 10.1016/j.fertnstert.2018.01.013. PMID: 29566852. In case the reader is interested in medical therapies with explanation and way of action, then can directly read the whole interesting paper by Vannuccini et al.
The MS has been completely revised to be methodologically correct. The conclusions, aim and limitations of the narrative review have also now been mentioned.
The editors accepted to publish a review paper with the above mentioned title from the first point.
Best regards and thanks a lot in advance
Georgios Kolovos

Reviewer 3 Report
Comments and Suggestions for Authors
Abstract is not structured so the purpose of the work is not easy to understand. Proposal - to form the abstrakt structure. In the text - it is a well done analysis of the data from the presented sources with important pointed view onthe issue. Instead of conclusion we find appropriate call for targeted attention to the issue.
In the row 99 mistake - tipo: gyneco logical instad of gynecological
In References:
*we can find sources older than 25 years egg. No13- but OK can be considered a classical source
*major mistakes in citations - probably caused by system used for making the list. In many of them the year is missing - citations are not unified. In No 6 even th name of the author is missing
Author Response
Dear Sir or Madam
I would like to thank you for your comments on our manuscript. You are absolutely right about the incomplete methodological presentation of the abstract and the article, which you can now find updated. We carried out a narrative review, which was completed according to the following checklist ( https://gpm.amegroups.org/art cle/view/10.21037/gpm-20-57/rc). The MS has been completely revised to be methodologically correct.
Any typos have been corrected. The references have also been updated with a new citation system.
Best regards
Georgios Kolovos

Round 2
Reviewer 1 Report
Comments and Suggestions for Authors
Thank you for addressing my comments, I have one minor comment:
In the method section, the authors mentioned they included the article up to may 2024, this sentence needs more clarifications, please edit the following, toe address:
- The study time period
- Number of articles have been identified and the number of articles have been included for this study.
- If possible add the search strategy (the author mentioned they have used the Mesh term, please include the search strategy), you may find an example here: Aazami, A., Valek, R., Ponce, A. N., & Zare, H. (2023). Risk and protective factors and interventions for reducing juvenile delinquency: A systematic review. Social Sciences, 12(9), 474.
We conducted a thorough literature search using PubMed, Scopus, and ScienceDirect databases, using MeSH or index terms for the keyword: “adenomyosis”. We included articles published in English language from MM/YYYY to May 2024, from a diverse range of sources such as original scientific papers, clinical trials, meta-analyses and reviews. We found xxxx articles, and xxx articles selected by the authors and additional references were also searched from the bibliographies of those chosen articles.
Author Response
Thanks a lot for your feedback and your response regarding the last minor changes. We added the following paragraph as recommended;
We included articles published in English language from the last thirty years, from May 1994 to May 2024, from a diverse range of sources such as original scientific papers, clinical trials, meta-analyses and reviews. We found 3625 articles from PubMed, 2170 from Scopus and 194 from Cochrane, and 43 articles selected by the authors and additional references were also searched from the bibliographies of those chosen articles.
Because of the broad topic of the article, covering many different issues, we were very open in the selection of articles that were reviewed and cited. Not like a narrative review with a very specific topic such as adenomyosis and surgical treatment. This is the reason why we did not add a PRISMA flowchart.
Best regards